# Comparative Transcriptome Analyses between Resistant and Susceptible Varieties in Response to Soybean Mosaic Virus Infection

**Yuanyuan Chen** †, **Ying Shen** †, **Boyu Chen, Lijun Xie, Yanmin Xiao, Zheng Chong, Han Cai, Guangnan Xing, Haijian Zhi and Kai Li** *

Soybean Research Institute, MARA National Center for Soybean Improvement, MARA Key Laboratory of Biology and Genetic Improvement of Soybean (General), State Key Laboratory for Crop Genetics and Germplasm Enhancement, Nanjing Agricultural University, Nanjing 210095, China; 2019801200@njau.edu.cn (Y.C.); 2019101088@njau.edu.cn (Y.S.); 2020801213@njau.edu.cn (B.C.); 2020101098@njau.edu.cn (L.X.); 2021101097@njau.edu.cn (Y.X.); 2021801225@njau.edu.cn (Z.C.); 2018101079@njau.edu.cn (H.C.); xinggn@njau.edu.cn (G.X.); zhj@njau.edu.cn (H.Z.)
* Correspondence: kail@njau.edu.cn
† The authors contributed equally to this work.

**Abstract:** Soybean mosaic virus (SMV) is a worldwide and hardly controlled virus disease in soybean. Kefeng-1 is an elite variety resistant to SMV in China. In order to discover resistance genes and regulation networks in Kefeng-1, we analyzed transcriptome data of resistant (Kefeng-1) and susceptible (NN1138-2) soybean varieties in response to infection of the SMV strain SC18 at 0, 6, and 48 hours post-inoculation (hpi) and 5 days post-inoculation (dpi). Many differentially expressed genes (DEGs) were identified with Kefeng-1 and NN 1138-2. Based on the enrichment analysis for gene ontology (GO) and the Kyoto Encyclopedia of Genes and Genomes (KEGG) pathway, we found that 48 hpi was the best time point for the defense response of the two soybean varieties in response to the SMV infection. The expression of seven candidate genes was further verified by qRT-PCR and was relatively consistent with the results of RNA-Seq. The expression of genes for *Glyma.11G239000* and *Glyma.18G018400*, members of the ethylene-insensitive 3/ethylene-insensitive3-like (EIN3/EIL) protein family involved in ETH, were downregulated in NN1138-2 but not in Kefeng-1 and the expression of *Glyma.14G041500* was upregulated in Kefeng-1 at 5 dpi. The expression of jasmonic acid repressor genes (TIFY/JAZ) was downregulated in NN1138-2 but not in Kefeng-1. NPR1 involved in the salicylic acid signaling pathway was downregulated in NN1138-2 at 48 hpi but upregulated in Kefeng-1. It shows that ethylene, jasmonic acid, and salicylic acid signaling pathways may be involved in the disease resistance process to the SMV strain SC18. Our findings would help to understand the molecular mechanism of soybean resistance to SMV.

**Keywords:** soybean; transcriptome; *Soybean mosaic virus*; plant hormone signal transduction; qRT-PCR



## 1. Introduction

Soybean (*Glycine max* (L.) Merr.) is an important food and feed crop in China with its rich nutritional value [1]. *Soybean mosaic virus* (SMV), a member of the genus *Potyvirus*, is the major pathogen causing soybean mosaic disease [2]. It can seriously affect the production and quality of soybean worldwide. According to the response to the eight different resistance soybean genotypes, seven SMV strains (G1-G7) were classified from 98 isolates in the United States [3]. Similarly, Japan reported 5 strains of SMV (A-E) and Korea reported 11 strains of SMV (G1-G7, SMV-N, G5H, G7A, and G7H) [4,5]. In China, the MARA National Center for Soybean Improvement collected more than 4500 SMV isolates nationwide and divided them into 22 strains (SC1-SC22) based on their responses to the 10 soybean identification hosts [6–9].

In the USA, three SMV resistance loci (*Rsv1*, *Rsv3,* and *Rsv4*) have been reported located on chromosomes 13, 14, and 2, respectively [10–13]. *Rsv1* showed resistance to SMV strains G1-G6 [14], *Rsv3* showed resistance to the virulent strains G5-G7 [12], and *Rsv4* showed resistance to most North American strains, except that strains G1 and G2 showed late susceptibility [15–19]. In China, resistance loci of the '$R_{SC}$' series were located on chromosomes 2, 13, 14, and 6 of the corresponding variety Kefeng-1, Qihuang-1, Dabaiba, and RN-9 [20–23]. Exploring resistance genes, breeding resistant materials, and studying resistant mechanisms is the most economical and effective method to control SMV disease.

With the rapid development of next-generation sequencing technology, RNA sequencing (RNA-Seq) has become an effective and comprehensive approach to explore the molecular mechanism of plants infected with a fungus or a virus [24–29]. The resistance gene of soybean to grey leaf spot (GLS) had been identified through transcriptomic and proteomic analyses combined with QTL region comparison, which is helpful to understand the molecular mechanisms of soybean to GLS resistance [30]. The relative expression of genes of the resistance and susceptible lines of Qihuang-1×NN 1138-2 was profiled and compared with the expression of three stages. Abscisic acid-induced genes (*PP2C3a*), three genes encoding calmodulin-like protein (*Glyma.03g28650*, *Glyma.19g31395,* and *Glyma.11g33790*), and JA signaling pathway related genes (*Glyma.01g41290* and *Glyma.11g04130*) were important for *Rsc3Q*-mediated resistance to SMV strain SC3 [31].

Kefeng-1 is a variety of broad-spectrum resistance to SMV. It carries a resistance gene (*R$_{SC18}$*) to SMV strain SC18, which is a weakly virulent strain only infecting NN 1138-2 and 8101 among the 10 soybean identification hosts. However, its resistance mechanism is not clear. Next generation sequencing has become the first choice to study the disease resistance mechanism of soybean to SMV, which reduces the sequencing cost and time. Thus, the purpose of this study was as follows: (i) to perform a global survey of the transcriptome expressions of Kefeng-1 (R) and NN1138-2 (S) inoculated with SMV strain SC18, (ii) to compare the transcriptional difference between Kefeng-1and NN1138-2, and (iii) to explore the disease resistance candidate genes for further functional genomic research in soybean.

## 2. Materials and Methods

### 2.1. Plant Materials and Virus Inoculation

Two parents, Kefeng-1 carrying a resistance gene to SC18 (*R$_{SC18}$*) and NN1138-2 carrying a susceptible gene to SC18 (*r$_{SC18}$*), were grown in a growth chamber at 26 ± 2 °C for 16-h light/8-h dark cycle. Three replicate (pots) of 10–15 seedlings per 10 cm diameter pot were grown in nutrient soil. When the first pair of leaves was fully expanded, the leaves were inoculated by a mechanical inoculation method [32]. The experimental group was inoculated with the SMV strain SC18 and the control group was inoculated with a phosphate buffer (PBS, 0.01 M, pH 7.4). After inoculation, the upper trifoliate leaves were collected at 0, 6, and 48 h post-inoculation and 5 days post-inoculation (each sample was repeated three times). All samples collected at each time point were frozen immediately in liquid nitrogen for further use.

### 2.2. RNA Extraction, Library Construction, and Sequencing

Total RNA was extracted using a Trizol reagent kit (Invitrogen, Carlsbad, CA, USA) according to the manufacturer's protocol. RNA quality was assessed on an Agilent 2100 Bioanalyzer (Agilent Technologies, Palo Alto, CA, USA) and checked using RNase free agarose gel electrophoresis. After total RNA was extracted, eukaryotic mRNA was enriched by Oligo (dT) beads, while prokaryotic mRNA was enriched by removing rRNA by a Ribo-ZeroTM magnetic kit (Epicentre, Madison, WI, USA). Then, the mRNA was cut into short fragments in a fragment buffer and reverse transcribed into cDNA with random primers. The second strand cDNA was synthesized with DNA polymerase I, RNase H, dNTP, and a buffer. Then, the cDNA fragments were purified with a QiaQuick PCR extraction kit (Qiagen, Venlo, The Netherlands), end repaired, poly (A) added, and ligated to Illumina sequencing adapters. The products were size selected by agarose gel electrophoresis, PCR

amplified, and sequenced using Illumina HiSeq2500 by Gene Denovo Biotechnology Co. (Guangzhou, China).

### 2.3. Read Alignment and Differential Expression Gene Screening

The reads obtained from the sequencer include raw reads containing adapters or low-quality bases that will affect the following assembly and analysis. In order to obtain high-quality clean reads, the reads were further filtered by fastp (version 0.18.0) [33]; the parameters were as follows: removing the reads containing adapters; removing the reads containing more than 10% unknown nucleotides (N); removing low-quality reads containing more than 50% of low quality (Q-value $\leq$ 20) bases. The paired-end clean reads were aligned to the reference genome of Glycine_max-Wm82.a2.v1 using HISAT (version 2.2.4, Daehwan Kim, Baltimore, MD, USA) [34]. The reads mapped to the exon region were also counted. Using the reference-based approach, the mapped reads of each sample were assembled using String Tie (version 1.3.1, Mihaela Pertea, Baltimore, MD, USA) [35,36]. An FPKM (transcripts per kilobase fragment/million mapped reads) method was used to analyze gene expression abundance and changes. The FPKM formula is shown as follows:

$$\text{FPKM} = \frac{10^6 \text{C}}{\text{NL}/10^3}$$

The expression of the FPKM (A) gene is: C to be the number of fragments that mapped to the gene A, N to be the total number of fragments that mapped to the reference genes, and L to be the number of bases on the gene A. The FPKM method is able to eliminate the influence of different gene lengths and the sequencing data amount on the calculation of the gene expression. Difference analysis is a statistical analysis of the differences in the gene expression between groups. The input data are the reads counts obtained from the gene expression level analysis, which is analyzed by edgeR and DESeq2 software. [37,38]. The genes/transcripts with the parameter of a false discovery rate (FDR) below 0.05 and an absolute fold change $\geq$2 were considered differentially expressed genes/transcripts.

### 2.4. Functional Enrichment and Pathway Analysis of DEGs

The latest genomic reference information of *Glycine max* was obtained from the Soybase (https://soybase.org/gb2/gbrowse/gmax2.0/, 24 July 2018), including Gene Ontology (GO) annotations for each gene. The Kyoto Encyclopedia of Genes and Genomes (KEGG) annotations were obtained from the KEGG database. A hypergeometric test was used to find out the GO terms and KEGG pathways that were significantly enriched by DEGs. The enrichment analyses of GO and KEGG were performed using the Omicshare online website (www.omicshare.com/tools, accessed on 14 October 2020).

### 2.5. RNA Extraction and Gene Expression Analysis by qRT-PCR

To verify the accuracy and reproducibility of the RNA-Seq data, qRT-PCR assays were conducted with gene specific primers. Total RNA from the same treated samples was extracted. Reverse transcription and qRT-PCR were performed using a PrimeScript™ RT reagent kit and an SYBR® Premix Ex Taq™ II kit (Nanjing, China), respectively. Each treatment contained three independent biological replicates and each experiment was repeated three times. The expression level of soybean *β-actin* gene was used as an internal reference. The fold change value of the gene expression was calculated using the $2^{-\Delta\Delta Ct}$ method. The sequences of specific primers were listed in Table S1. Briefly, the qRT-PCR primer pairs of each gene were designed using Primer Premier 5.0 (PREMIER Biosoft International, Palo Alto, CA, USA) according to the following criteria: the melting temperature between 54 °C and 62 °C, a primer length between 18 bp and 25 bp, and an amplified product length of 100–200 bp. PCR volume was 20 µL containing 1.0 µL of cDNAs, 1.0 µL of forward and reverse primers, 10.0 µL SYBR Green PCR master mix (Applied Biosystems, Shanghai, China), and 7 µL ddH₂O. qRT-PCR analysis was carried out on an Eppendorf Mastercycler

Realplex 2.2 detection system according to the following profile: denaturation at 95 °C for 3 min, followed by 40 cycles at 95 °C for 15 s, 58 °C for 30 s, and then 72 °C for 30 s.

## 3. Results

### 3.1. Phenotypic Identification of NN1138-2 and Kefeng-1 after Inoculation with SC18

Fifteen days after inoculation with SC18, the susceptible variety NN 1138-2 showed a shrinkage of the mosaic leaves (Figure 1A), while the leaf phenotype of the resistant variety Kefeng-1 had no significant difference compared with the control inoculated with PBS; no symptom of mosaic and shrinkage appeared (Figure 1B).

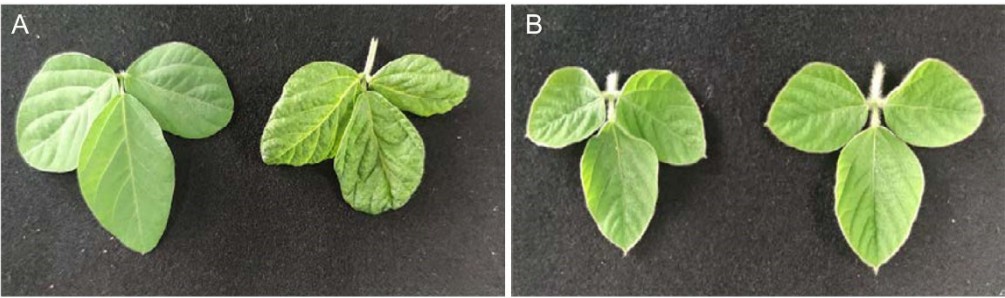

**Figure 1.** Phenotype of NN 1138-2 and Kefeng-1 inoculated with PBS (left) and SC18 (right) fifteen days later. (**A**): NN 1138-2; (**B**): Kefeng-1.

### 3.2. Evaluation of RNA-Seq Data

In order to study the changes in the gene expression levels of Kefeng-1 (R) and NN 1138-2 (S) at 0, 6, 48 h, and 5 day post inoculation with SC18, a transcriptome sequencing experiment was conducted. Due to the high-quality requirements of RNA sequencing, the quality and the concentration of RNA are strictly tested. The detection results are shown in Table S2: the RNA integrity value was around seven, and the lowest concentration was 209 ng/μL and the highest concentration was 1089 ng/μL. The quality and concentration met the requirements of sequencing. A total of 24 independent cDNA libraries were obtained. The size of the cDNA library was 300–400 bp. The screening and the quality testing of raw data are shown in Table S3. The Q30 percentages of the raw data were all above 92.32%. The number of clean reads ranged from 36,076,218 bp to 52,528,690 bp and the percentage of clean data is greater than 99% (Table S3). The reads total mapped on the reference genome was more than 93% and the uniquely mapped reads were over 88% (Table S4). The statistical data of genomic region location showed that the proportion of mapping to exon region locations exceeds 91% (Table S5). These data indicated that reliable transcriptome data can be used for subsequent differential gene analysis. The raw data from RNA-sequencing were deposited in the public database NCBI (accession number: PRJNA861407) (https://www.ncbi.nlm.nih.gov/sra/PRJNA861407, accessed on 25 July 2022).

### 3.3. Statistics on the Number of DEGs

DEGs were screened out between the treatments with $\log_2 FC$ (fold change) >1 and *p*-adj (adjusted *p*-value) < 0.05. Compared with the control (0 hpi), there were 9995 differential gene expressions at the early stages of infection (6 hpi) in the susceptible variety NN1138-2 after SC18 inoculation, including 6368 upregulated genes and 3627 downregulated genes. At 48 hpi, there were a total of 6938 differential gene expressions, of which 3302 were upregulated and 3636 were downregulated. There were 12,512 differential expressed genes at 5 dpi, including 7314 upregulated genes and 5198 downregulated genes (Figure 2A). A total of 2307 genes were differentially expressed throughout the infection process (Figure 2B). In the resistant variety Kefeng-1, there were 9227 differential expressed genes at the early stages of infection (6 hpi), including 6175 upregulated genes and 3052 downregulated genes. There were 7357 differential expressed genes at the late stages

of infection (48 hpi), including 4269 upregulated genes and 3088 downregulated genes. At 5 dpi, there were a total of 11,014 differential expressed genes, of which 6397 were upregulated and 4617 were downregulated (Figure 2A). The expression trends of the differential genes throughout the time point were similar to those of the susceptible variety NN1138-2, and there were a total of 2443 DEGs throughout the infection process (Figure 2C).

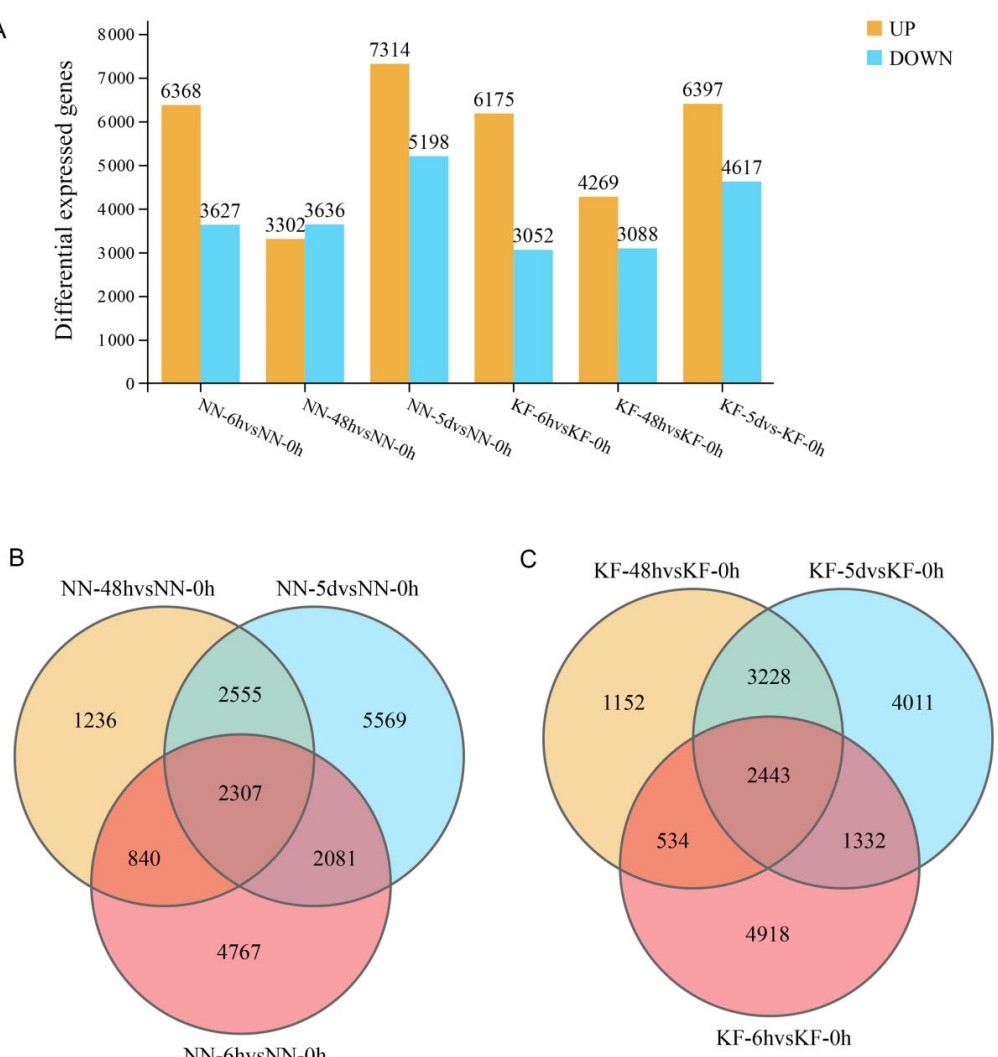

**Figure 2.** Differentially expressed genes (DEGs). (**A**): analysis of differential gene expression in NN 1138-2 and Kefeng-1. (**B**): analysis of common differential gene expression at different treatment times in NN 1138-2. (**C**): analysis of common differential gene expression at different treatment times in Kefeng-1.

*3.4. Functional Annotation and Pathway Enrichment Analysis of DEGs*

Gene Ontology (GO) enrichment analysis was used to determine the functional classification of DEGs between different treatments. Genes were divided into three categories: biological process, molecular function, and cellular component. Among them, the enrichment items of biological process were the most common. Compared with 0 hpi, the GO functional classifications of Kefeng-1 and NN1138-2 were similar throughout the infection processes (6 hpi, 48 hpi, and 5 dpi) after inoculation. In biological processes, it is mainly enriched in metabolic processes, cellular processes, and single biological processes. In terms of molecular functions, DEGs were mostly enriched in merging processes and catalytic activity. In cell components, DEGs of susceptible and resistant varieties were mostly enriched in cells and organelles at the early stage of infection (6 hpi) and in membranes and cells at the late stages of infection (48 hpi and 5 dpi) (Figure 3). It is worth noting that, compared

with 0 hpi, at 6 hpi and 5 dpi, the DEGs with upregulated were more than downregulated in the susceptible and resistant varieties enriched in the GO pathway (Figure 3A,C,D,F), while at 48 hpi, the increase of downregulated genes was higher than that of upregulated genes in the susceptible varieties (Figure 3B), and the upregulated genes was still more than the downregulated genes in the resistant varieties (Figure 3E).

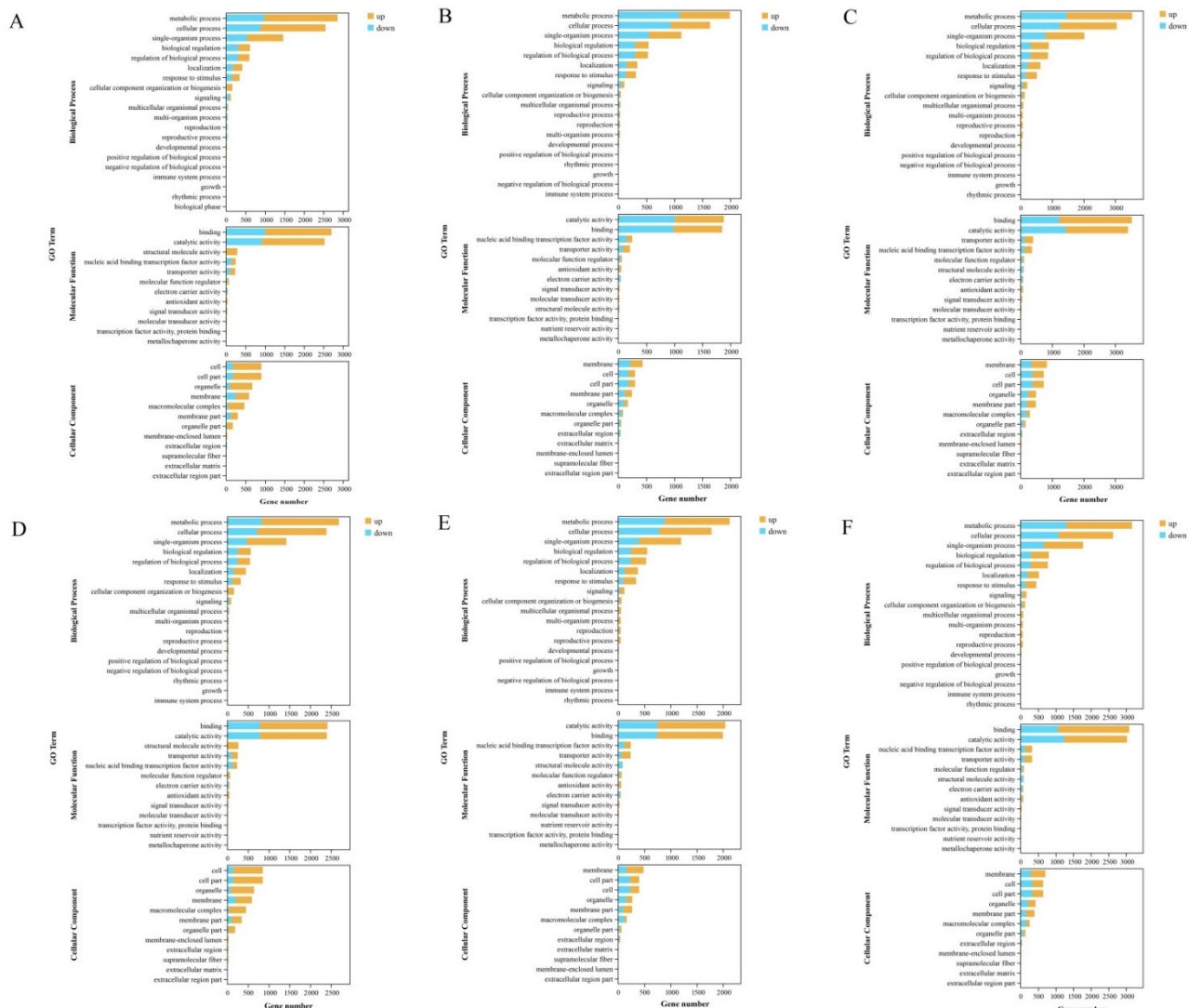

**Figure 3.** Gene Ontology (GO) function enrichment analysis of DEGs identified. (**A**) GO analysis of DEGs between 6 hpi vs. 0 hpi in susceptible variety. (**B**) GO analysis of DEGs between 48 hpi vs. 0 hpi in susceptible variety. (**C**) GO analysis of DEGs between 5 dpi vs. 0 hpi in susceptible variety. (**D**) GO analysis of DEGs between 6 hpi vs. 0 hpi in resistant variety. (**E**) GO analysis of DEGs between 48 hpi vs. 0 hpi in resistant variety. (**F**) GO analysis of DEGs between 5 dpi vs. 0 hpi in resistant variety.

The Kyoto Encyclopedia of Genes and Genomes (KEGG) enrichment analysis of DEGs was performed to determine changes in metabolic pathways following the viral infection. Figure 4 shows the top 10 enrichment pathways (*p*-value < 0.001) for each group of DEGs. The ribosome was the most significantly enriched in susceptible and resistant varieties at the early stage of infection (6 hpi) and plant–pathogen interaction was the most significantly enriched in the susceptible variety and phenylpropanoid biosynthesis in the resistant variety at the late stage of infection (48 hpi). The most significant enrichment in the susceptible and resistant varieties was the biosynthesis of secondary metabolites (Figure 4). Compared with 0 hpi, the plant–pathogen interaction pathways were significantly enriched

in the susceptible variety throughout the infection processes (6 hpi, 48 hpi, and 5 dpi) after inoculation (Figure 4), while the plant–pathogen interaction pathways were significantly enriched at the late stage of infection (48 hpi) in the resistant variety. The MAPK (mitogen-activated protein kinase) signal transduction pathway is composed of highly conserved serine/threonine protein kinases MAPKKK, MAPKK, and MAPK, which play an important role in plant-resisting biotic and abiotic stresses. This study found that both NN1138-2 and Kefeng-1 were significantly enriched at 6 hpi, 48 hpi, and 5 dpi compared with 0 hpi, indicating that this pathway has a better effect on soybean after SMV inoculation. In addition, related plant-defense pathways, such as secondary metabolites biosynthesis, plant hormone signal transduction pathways, isoflavonoid metabolic pathways, and phenylpropanoid biosynthesis pathways play an important role in stress resistance. We found that these pathways were enriched significantly at the late stage of infection (48 hpi) both in NN1138-2 and Kefeng-1 (Figure 4). Therefore, combining GO and KEGG enrichment analyses, it was speculated that the late stage of infection (especially 48 hpi) may provide some help for studying the different defense responses of the susceptible varieties (NN1138-2) and the resistant varieties (Kefeng-1) to soybean mosaic virus infection.

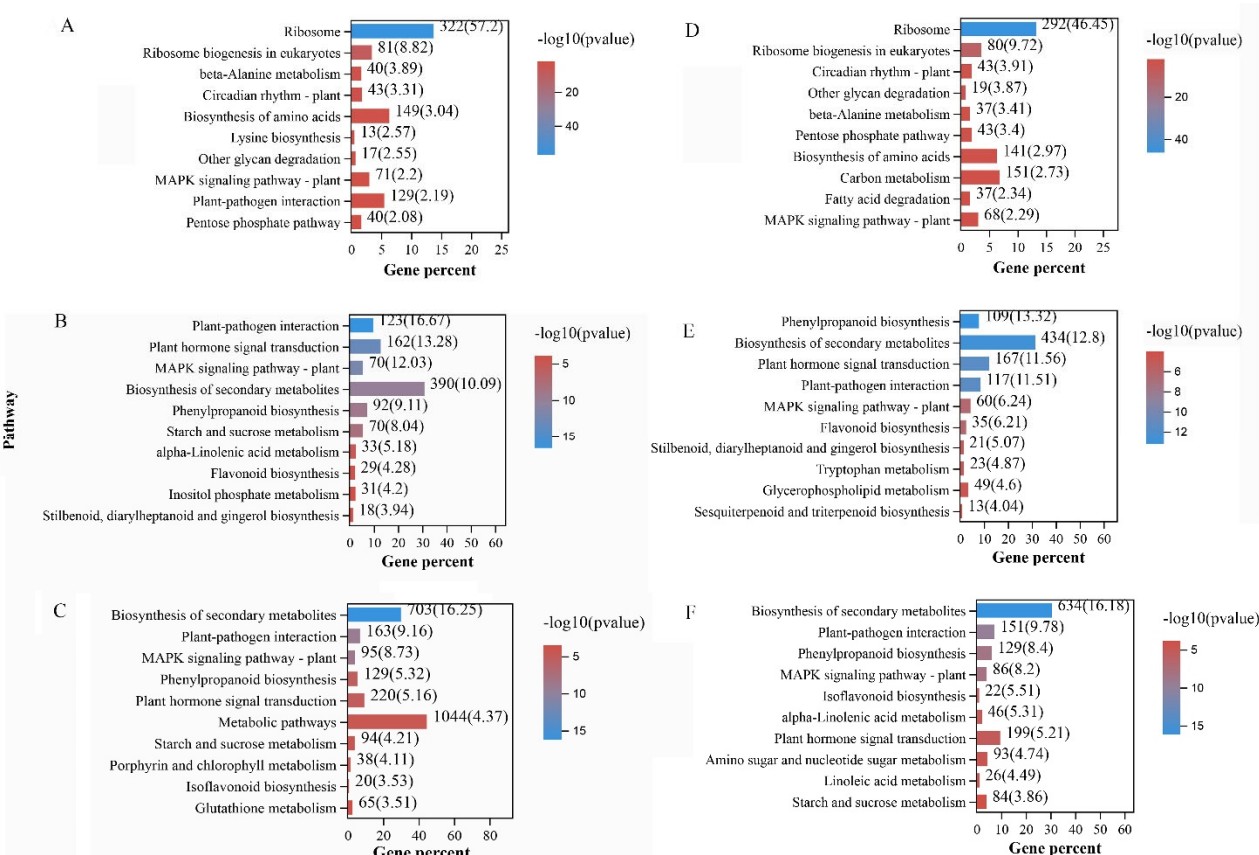

**Figure 4.** Kyoto Encyclopedia of Genes and Genomes (KEGG) enrichment analysis of DEGs identified. (**A**) KEGG analysis of DEGs between 6 hpi vs. 0 hpi in susceptible variety. (**B**) KEGG analysis of DEGs between 48 hpi vs. 0 hpi in susceptible variety. (**C**) KEGG analysis of DEGs between 5 dpi vs. 0 hpi in susceptible variety. (**D**) KEGG analysis of DEGs between 6 hpi vs. 0 hpi in resistant variety. (**E**) KEGG analysis of DEGs between 48 hpi vs. 0 hpi in resistant variety. (**F**) KEGG analysis of DEGs between 5 dpi vs. 0 hpi in resistant variety.

## 3.5. Plant Hormone Effects on Soybean Defense against SMV

Plant hormone signal transduction pathways play an important role in soybean defense against the SMV strain SC18 and were significantly enriched in NN1138-2 and Kefeng-1 at 48 hpi (Figure 4). The analysis of the plant hormone signal transduction pathway showed that, compared with 0 hpi, ethylene pathway-related genes in NN 1138-2

were downregulated at 48 hpi and 5 dpi, while upregulated in Kefeng-1 (Figures S1–S4). Two DEGs in the ethylene (ET), encoding ethylene-insensitive3/ethylene-insensitive3-like (EIN3/EIL) protein family (*Glyma.11G239000*, *Glyma.18G018400*), were downregulated in NN1138-2 and *Glyma.14G041500* was upregulated in Kefeng-1 at 5 dpi (Table 1). At 48 hpi, in the jasmonic acid pathway, compared with 0 hpi, JAZ and MYC2 signaling pathways were inhibited in NN1138-2 (Figure S1), while the JAZ signaling pathway was inhibited in Kefeng-1, but some MYC2-related DEGs were upregulated (Figure S3). *Glyma.01G096600* and *Glyma.07G051500* are MYC2 transcription factors, which were downregulated in NN1138-2 at 48 hpi. *Glyma.01G096600* was downregulated and *Glyma.14G117300* related to MYC2 was upregulated in Kefeng-1 at 48 hpi (Table 1). At 5 dpi, the JAZ signaling pathway was still suppressed in Kefeng-1 (*Glyma.09G077500* and *Glyma.17G043700* were downregulated) and *Glyma.14G117300* related to MYC2 was upregulated in NN1138-2, while one DEG (*Glyma.13G116100*) related to the JAZ signaling pathway was upregulated in Kefeng-1, and some MYC2-related DEGs (*Glyma07G051500* and *Glyma.14117300*) were upregulated (Figures S2 and S4 and Table 1). NPR1 plays an important role in SA-responsive induced systemic resistance and is a major regulator of plant systemic acquired resistance (SAR). Compared with 0 hpi, DEGs related to NPR1 showed different expression trends at the late stage of infection; downregulated in NN1138-2 at 48 hpi and upregulated at 5 dpi, while upregulated in Kefeng-1 all the time (Figures S1–S4). All of the above indicated that ETH, JA, and SA signaling pathways were involved in the soybean defense responses to SMV SC18 infection. These results also reflected the different defense abilities of the susceptible variety (NN1138-2) and the resistant variety (Kefeng-1) against SC18 infection and the differences in transcription level. The resistant variety (Kefeng-1) had a faster defense against SMV infection.

### 3.6. Validation of RNA-Seq Data by qRT-PCR

In order to further confirm the gene expression pattern obtained from RNA-Seq, seven DEGs were selected for qRT-PCR, including protein PP2C37 (*Glyma.03G219000*, *Glyma.11G222600*, *Glyma.18G035000*), AS1 (*Glyma.18G061100*), *Glyma.09G238300*, *Glyma.12G183400*, and *Glyma.13G267800* (Table S6). According to the results of RNA-Seq, there were two genes, *Glyma.09G238300* and *Glyma.13G267800*, that responded positively at early infected times (0 and 6 hpi) in NN 1138-2. There were two genes (*Glyma.03G219000* and *Glyma.12G183400*) that responded at late infected times (48 hpi and 5 dpi), and three genes (*Glyma.11G222600*, *Glyma.18G035000*, and *Glyma.18G061100*) that responded throughout all periods in NN 1138-2 (Figure 5). In Kefeng-1, there were two genes (*Glyma.09G238300* and *Glyma.13G267800*) that responded positively at early infected times (0 and 6 hpi) and three genes (*Glyma.11G222600*, *Glyma.18G035000*, and *Glyma.18G061100*) that responded at late infected times (48 hpi and 5 dpi). There were two genes (*Glyma.03G219000* and *Glyma.12G183400*) that responded throughout all periods (Figure 6). RNA-Seq results showed that at 6–48 hpi, the expression levels of *Glyma.03G219000*, *Glyma.11G226000*, *Glyma.13G267800*, and *Glyma.18G061100* downregulated in the susceptible variety and upregulated in the resistant variety (Figure S5). The expression of *Glyma.12G183400* was significantly different among the resistant and susceptible varieties, with the expression level first downregulating and then upregulating in the susceptible variety, while reversed in the resistant variety. (Figure S5). *Glyma.18G035000* was upregulated in the susceptible variety and downregulated in the resistant variety (Figure S5) at early infected times (0 and 6 hpi). In the susceptible variety, the qRT-PCR and RNA-Seq expression patterns of *Glyma.03G219000* were different at early infected times (0 and 6 hpi) and the expression patterns of *Glyma.11G222600* and *Glyma.12G183400* were different at late infected times (48 hpi to 5 dpi) (Figure 5). *Glyma.03G219000* and *Glyma.11G222600* showed different expression patterns at early infected times (0–6 hpi) in the resistant variety (Figure 6). Although the expression patterns of qRT-PCR and RNA-Seq were inconsistent in certain time periods, the correlation coefficient between qRT-PCR and RNA-Seq was greater than 0.9, indicating significant correlation. The correlation coefficient of some genes was around 0.8, indicating a good correlation. The results revealed that RNA-Seq were reliable.

**Table 1.** DEGs involved three plant hormone signal pathways (SA, JA, and ETH) between susceptible and resistant varieties after SMV inoculation.

| Gene ID | Description | Symbol | log$_2$FC (NN-48 h/0 h) | log$_2$FC (NN-5d/0 h) | log$_2$FC (KF-48 h/0 h) | log$_2$FC (KF-5d/0 h) | Pathway |
|---|---|---|---|---|---|---|---|
| *Glyma.11G239000* | putative ETHYLENE INSENSITIVE 3-like 4 protein (Glycine max(soybean)) | EIN/EIL | −7.4 | −7.4 | - | - | ETH |
| *Glyma.14G041500* | ETHYLENE INSENSITIVE 3-like 1 protein (Glycine max(soybean)) | | - | - | - | 1.8 | |
| *Glyma.18G018400* | putative ETHYLENE INSENSITIVE 3-like 4 protein (Glycine max(soybean)) | | - | −4.5 | - | - | |
| *MSTRG.38091* | the new gene | ETR | 2.5 | 2.5 | - | - | |
| *MSTRG.38100* | the new gene | | −2.4 | - | - | - | |
| *Glyma.09G002600* | ethylene receptor (Glycine max(soybean)) | | 1.1 | 2 | - | 1.1 | |
| *Glyma.12G241700* | ethylene receptor (Glycine max(soybean)) | | 1.2 | - | - | - | |
| *Glyma.19G213300* | ethylene response sensor 1 (Glycine max(soybean)) | | 1.1 | - | - | - | |
| *Glyma.09G077500* | uncharacterized LOC100306045 (Glycine max(soybean)) | TIFY/JAZ | - | −1.1 | - | - | JA |
| *Glyma.13G116100* | CCT motif and tify domain containing protein (Glycine max(soybean)) | | - | - | - | 1.4 | |
| *Glyma.17G043700* | CCT and tify domains containing protein (Glycine max(soybean)) | | −1.1 | −1.7 | - | - | |
| *Glyma.01G096600* | transcription factor MYC2 (Glycine max(soybean)) | MYC2 | −1.3 | 11.1 | 11.4 | - | |
| *Glyma.07G051500* | transcription factor MYC2 (Glycine max(soybean)) | | −2 | - | - | 1.4 | |
| *Glyma.14G117300* | uncharacterized LOC100775685 (Glycine max(soybean)) | | - | 2.4 | 1.2 | 2.9 | |
| *Glyma.03G128600* | BTB/POZ domain and ankyrin repeat containing protein COCH | NPR1 | - | 1.8 | 1.3 | - | SA |
| *Glyma.09G020800* | NPR1-1 protein (Glycine max(soybean)) | | −1 | - | - | - | |
| *Glyma.14G031300* | BTB/POZ domain and ankyrin repeat containing protein NPR1 (Glycine max(soybean)) | | - | - | - | 1.1 | |
| *Glyma.19G131000* | regulatory protein NPR5-like (Glycine max(soybean)) | | - | 2.5 | - | - | |

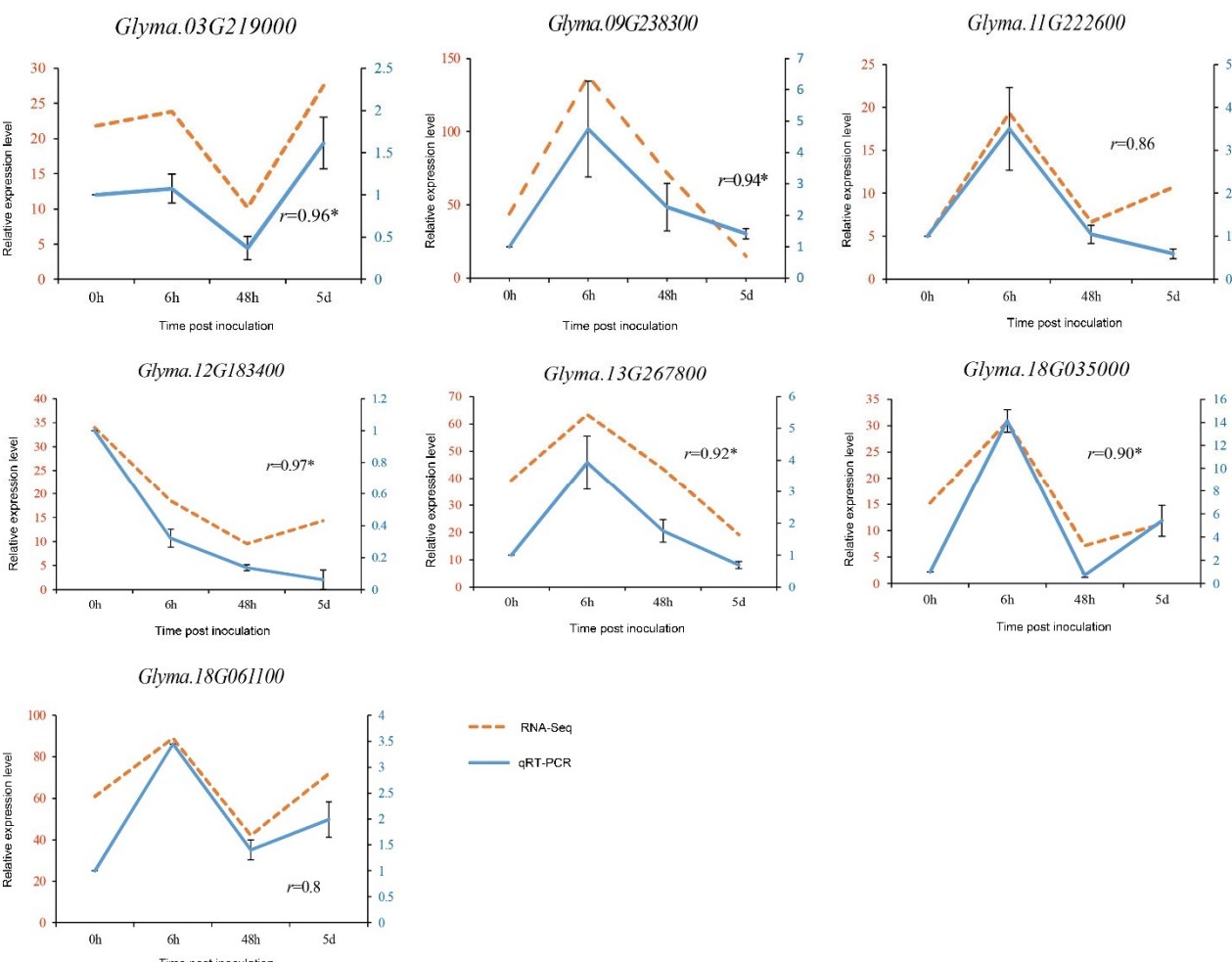

**Figure 5.** Comparison of the relative expression level change of 7 selected DEGs by qRT-PCR and RNA-Seq (susceptible variety NN1138-2, left vertical axis coordinate is FPKM of RNA-Seq (orange); right vertical axis coordinate is relative expression level of qRT-PCR (blue); *r*-values are the correlation coefficients between qRT-PCR and RNA-seq, * indicates significant correlation).

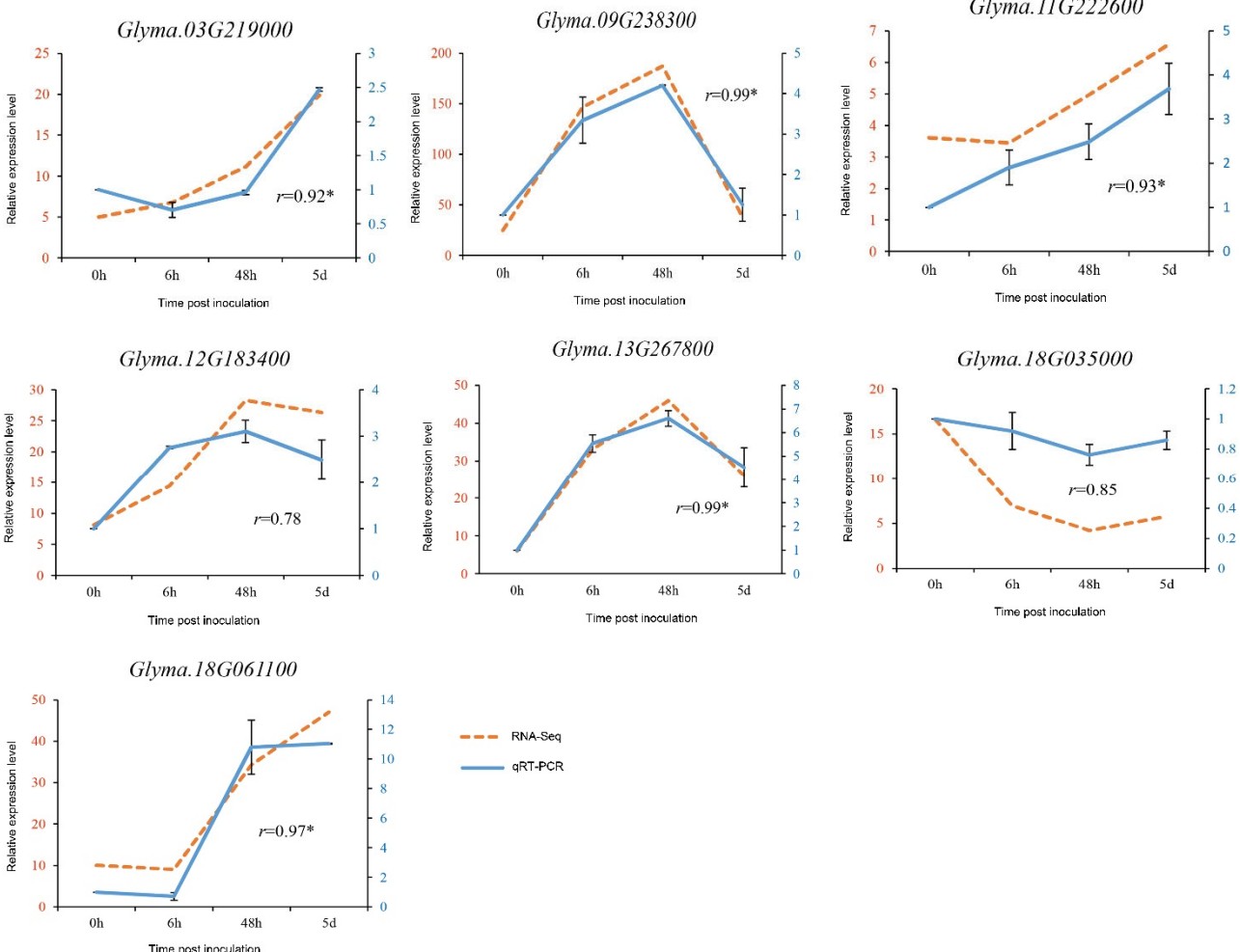

**Figure 6.** Comparison of the relative expression level change of 7 selected DEGs by qRT-PCR and RNA-Seq (resistant variety Kefeng-1, left vertical axis coordinate is FPKM of RNA-Seq (orange); right vertical axis coordinate is relative expression level of qRT-PCR (blue); *r*-values are the correlation coefficients between qRT-PCR and RNA-seq, * indicates significant correlation.

## 4. Discussion

Kefeng-1 has broad-spectrum resistance to *soybean mosaic virus*. Although the SMV strain SC18 is less virulent, it can only infect NN1138-2 and 8101 among the 10 identified hosts, but it spreads widely and occurs in major producing regions of southern and northeastern China [1,8]. Therefore, it is particularly important to explore resistance genes of this strain for promoting SMV resistance breeding.

RNA-Seq is a common analytical tool in biology that can be used to analyze the expression levels of specific genes, discover candidate genes associated with specific phenotypes or disease resistance, develop SSR markers, and explore host-pathogen interactions [39]. The transcriptome is the total amount of RNA of a specific cell, tissue, or organism at a particular time or in a particular functional state [40,41]. After inoculation of SMV, 9995, 6368, and 12,512 DEGs were identified in the susceptible variety at 6, 48 hpi, and 5 dpi, and 9227, 7357, and 11,014 DEGs were identified in the resistant variety at 6, 48 hpi, and 5 dpi. Moreover, the susceptible variety (NN1138-2) showed more DEGs than those in the resistant variety (Kefeng-1) except for DEGs at 48 hpi (Figure 2A), which may be the reason for the disturbance of the normal metabolic system after inoculation of the susceptible variety with SMV.

According to the GO analyses in this study, compared with 0 hpi, the functional classification of the GO at 6, 48 hpi, and 5 dpi was similar for the susceptible variety and the resistant variety. In biological processes, it is mainly enriched in metabolic processes, cellular processes, and single biological processes. In terms of molecular functions, DEGs were mostly enriched in the merger process and the catalytic activity. In terms of cell components, DEGs of susceptible and resistant varieties were mostly enriched in cells and organelles at early infected times (6 hpi), and in membranes and cells at late infected times (48 hpi and 5 dpi) (Figure 3). The difference was observed in the susceptible variety at 48 hpi, where the number of downregulated DEGs was more than the upregulated DEGs (Figure 3B). Obviously, under the action of SMV, DEGs were mainly upregulated in the resistant variety at 48 hpi (Figure 3E), however the number of downregulated genes were increased in the susceptible variety at 48 hpi (Figure 3B). This may suggest that the resistant variety had a positive response to SMV at 48 hpi.

The Kyoto Encyclopedia of Genes and Genomes (KEGG) is equivalent to a reference knowledge base, which is based on the principle of linking genomes to life through the process of pathway mapping, that is, the genome or transcriptome content of genes mapped to a KEGG reference pathway to infer the systemic behavior of a cell or organism [42]. The MAPK (Mitogen-Activated Protein Kinase) signal transduction pathway plays an important role in plant resistance to biotic and abiotic stress [43]. In our study, it was found that both NN1138-2 and Kefeng-1 were significantly enriched at 6 hpi, 48 hpi, and 5 dpi compared with 0 hpi, indicating that this pathway played a good defense role in soybean after SMV inoculation (Figure 4). In addition, related to plant defense pathways, such as secondary metabolites biosynthesis, plant hormone signal transduction pathways, isoflavonoid metabolic pathways, and phenylpropanoid biosynthesis pathways play an important role in stress resistance. We found that in both the susceptible variety and the resistant variety, these pathways were enriched significantly at late infected times (48 hpi) (Figure 4).

Furthermore, we found that there are significant differences in the expression of ETH, JA, and SA related genes in the signal transduction pathway of plant hormones (Figures S1–S4). Plant hormones play an important role in the growth and development of plants, some of which are essential for plant immunity; salicylic acid (SA), jasmonic acid (JA), and ethylene (ETH) play a dominant role in disease resistance. *Glyma.11G239000* and *Glyma.18G018400* involved in ETH, from the ethylene-insensitive3/ethylene-insensitive3-like (EIN3/EIL) protein family, were downregulated in the susceptible variety and *Glyma.14G041500* was upregulated in the resistant variety at 5 dpi (Table 1). EIN3 is a key transcriptional regulator in ethylene signaling [44]. The ethylene-insensitive3/ethylene-insensitive3-like (EIN3/EIL) protein family functions as an important factor for plant growth and development under various environmental conditions [45]. Therefore, we hypothesized that the growth and development of the susceptible variety is more susceptible to SMV compared with the resistant variety.

The role of jasmonic acid (JA) in plant defense against viruses is controversial [46]. JA plays an important role in inducing resistance to pathogen infection and insect herbivory [47]. Arabidopsis mutants with impaired JA perception or biosynthesis are unable to deploy effective defense responses to pathogen infection [48–51]. In the present study, three DEGs (*Glyma.09G077500, Glyma.13G116100,* and *Glyma.17G043700*) coming from *Arabidopisis thaliana* tify family had different expressions in the susceptible and the resistant varieties (Table 1). Among them, *Glyma.09G077500* was downregulated at 5 dpi and *Glyma.17G043700* was downregulated at late infected times (48 hpi and 5 dpi) in the susceptible variety. *Glyma.13G116100* was upregulated at 5 dpi in the resistant variety (Table 1). The MYC2 transcription factor is a member of the plant BHLH family of transcription factors and it plays a central role in the plant JA signaling pathway [52]. *Glyma.01G096600*, which is an MYC2 transcription factor, was downregulated at late infected times (48 hpi and 5 dpi) in the susceptible variety, while only downregulated at 5 dpi in the resistant variety. *Glyma.07G051500* (MYC2) was downregulated at 48 hpi in the susceptible variety

and upregulated at 5 dpi in the resistant variety. *Glyma.14G117300* related to MYC2 was upregulated at 5 dpi in the susceptible variety and upregulated at both 48 hpi and 5 dpi in the resistant variety (Table 1). Therefore, we speculate that JA probably had a positive role in the resistance response to SMV, which was similar to previous reports [53,54]. The SA signaling pathway is the main defensive pathway in plant defense against viruses [46]. In our study, NPR1 in the SA signaling pathway was downregulated in the susceptible variety at 48 hpi and upregulated at 5 dpi, while upregulated in the resistant variety all the time (Figures S1–S4). *Glyma.09G020800*, encoding NPR1-1 protein, was downregulated at 48 hpi in the susceptible variety (Table 1). *Glyma.19G131000*, encoding NPR1 protein, was upregulated at 5 dpi in the resistant variety (Table 1). Luo et al., [55] transferred the *AtNPR1* gene into different rice lines and the transgenic lines showed higher resistance to rice leaf blight, with disease resistance increased by more than 90%. Therefore, we speculate that SA probably had a positive role in the resistance response to SMV.

Seven candidate genes that may be related to SC18 inoculation were screened by a Venn diagram, gene expression, and gene function analysis. These genes were verified by qRT-PCR and the expression trend of qRT-PCR and RNA-Seq were basically the same. The correlation between the two was more than 70% and most of them were more than 90%, showing significant correlation (Figures 5 and 6). These results indicated that RNA-Seq was more reliable. These genes have different expression patterns in resistant and susceptible varieties. Among them, three genes, *Glyma.03G219000, Glyma.11G222600,* and *Glyma.18G035000*, were associated with 2C-type serine/threonine protein phosphatases (PP2C), and they had the same expression pattern in the susceptible variety and the expression of PP2C-type protein phosphatases was higher in the susceptible variety than in the resistant variety (Figure S5). From Table S6, we could see that the pathways they involved in were the plant hormone signaling pathway and the MAPK signaling pathway. Plant hormones play an important role in plant growth and development and some of them are essential for plant immunity [56]. The MAPK signaling pathway (mitogen-activated protein kinase) can be involved not only in plant growth and development but also in the plant defense process [57]. *Glyma.18G061100* was related to asparagine synthetase and was involved in plant metabolic processes such as synthesis of secondary metabolites, aspartate, alanine, and glutamate metabolism (Table S6). Asparagine synthetase (AS) is associated with plant disease resistance and defense processes, and AS1 is one of the AS gene family. It has been shown that the AS1 gene has a regulatory role in nitrogen metabolism, and overexpression of AS1 gene enhances plant-transported nitrogen and also enhances seed accumulation of storage proteins [58]. *Glyma.03G219000, Glyma.11G222600, Glyma.18G035000,* and *Glyma.18G061100* were all expressed in the susceptible variety at higher levels than in the resistant variety (Figure S5) and may be genes that promote SMV infection. The composition of plant epidermal cuticles and waxes is associated with plant disease resistance. The epidermal wax layer is the first area of contact between plant pathogens and the host, and the activity of pathogens is influenced by the epidermal wax layer of the plant, which functions as a defense against pest and disease infestation [59,60]. *Glyma.12G183400* is associated with fatty acyl-CoA reductase (FAR) (Table S6), a key enzyme in the lipid synthesis pathway that catalyzes the direct reduction of fatty acyl-CoA to primary alcohols, which are further converted to waxes by the action of wax synthase. *Glyma.12G183400* expression patterns were completely opposite between the resistant and susceptible varieties, and *Glyma.12G183400* expression was upregulated at all time points in the resistant variety and downregulated at all time points in the susceptible variety compared with the control 0 hpi (Figure S5). Wen et al., [61] used a high-density custom single nucleotide polymorphism (SNP) array (52041 SNP) for genotypic analysis of two soybean diversity groups. Genome-wide association studies were conducted to identify quantitative trait loci (QTL) controlling resistance to white mold (*White mould*) in combination with resistance variation data observed in field and greenhouse environments, and the results showed that 16 and 11 loci were significantly associated with resistance in the field and greenhouse, respectively, and *Glyma.12G183400* was a candidate gene

belonging to one of these loci. In addition, it was reported previously that *Glyma.12G183400* was associated with the oxidative stress response [62]. The function of these genes should be further studied

Our transcriptome data provided a basis for a comprehensive understanding of the gene expression profiles of two different soybean resistance phenotypes at different stages of SMV infection and provided an important starting point for further understanding of the soybean mosaic virus resistance mechanism. We speculated that the signal transduction pathway of plant hormone has a certain influence on soybean disease resistance. The results highlighted the important genes involved in ETH, JA, and SA signaling pathways and related to response to SMV expressed differently in susceptible and resistant varieties. We will further focus on the molecular function analysis of these important genes during soybean and SMV interaction. The findings would help to clarify the molecular mechanisms of SMV resistance in soybean.

**Supplementary Materials:** The following supporting information can be downloaded at: https://www.mdpi.com/article/10.3390/agronomy12081785/s1, Figure S1: Ethylene, jasmonic acid and salicylic acid pathways in the phytohormone signaling pathway of Nannong 1138-2 at 48 hpi; Figure S2: Ethylene, jasmonic acid and salicylic acid pathways in the phytohormone signaling pathway of Nannong 1138-2 at 5 dpi; Figure S3: Ethylene, jasmonic acid and salicylic acid pathways in the phytohormone signaling pathway of Kefeng No.1 at 48 hpi; Figure S4: Ethylene, jasmonic acid and salicylic acid pathways in the phytohormone signaling pathway of Kefeng No.1 at 5 dpi; Figure S5: Expression of other possible candidate genes related to SMV inoculated; Table S1: Primers used for qRT-PCR validation; Table S2: The quality and concentration measurements of RNA sample; Table S3: Filtering and quality testing of raw data: Tables S4: Statistics of RNA-Seq map; Table S5: Comparison of regional statistics; Table S6: Function of candidate genes.

**Author Contributions:** Y.C. and K.L. designed the experiments and obtained funding for the research. Y.C. and Y.S. contributed to compiling and analyzing the data and wrote the manuscript. B.C. and G.X. conducted statistical analysis. Y.C., L.X., Y.X., Z.C. and H.C. performed the experimental analyses. H.Z. participated in the data analysis and supervised the writing of the manuscript. All authors have read and agreed to the published version of the manuscript.

**Funding:** This work was financially supported through grants from the National Key R&D Program of China (2021YFD1201604), the open competition project of seed industry revitalization of Jiangsu Province (JBGS [2021] 060), the National Natural Science Foundation of China (grant No. 31671718), and China Agriculture Research System of MOF and MARA (No. CARS-04), the Jiangsu Collaborative Innovation Center for Modern Crop Production (JCIC-MCP), Collaborative Innovation Center for Modern Crop Production co-sponsored by Province and Ministry (CIC-MCP).

**Conflicts of Interest:** The authors declare no conflict of interest.

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
