# Peer review of "Comparative Transcriptome Analyses between Resistant and Susceptible Varieties in Response to Soybean Mosaic Virus Infection"

_agronomy, doi:10.3390/agronomy12081785_

Round 1
Reviewer 1 Report
In this manuscript, authors conducted RNA-sequencing to identify differentially expressed genes between resistant and susceptible cultivars in response to soybean mosaic virus infection. In general, the purposes of the study and experimental approaches were fine. But there were several technical problems in this manuscript.
The first was identification of differentially expressed genes. Authors should use the number of mapped reads (read count) instead of FPKM values using DESeq2 or edgeR. Check the manual for two programs.
The second was that the font sizes in the figures were very small. Reanalyze the data and prepare high quality figures for the manuscript.
The third was that the manuscript was very poorly written. Lots of sentences were awkward. There were lots of mistakes. English was really very bad.
The fourth was that authors should provide accession numbers of raw data from RNA sequencing after depositing raw data in publica database such as SRA database.
My suggestions are as follows.
The title might be changed to “Comparative transcriptome analyses between resistant and susceptible varieties in response to soybean mosaic virus infection”.
L3 and L14 Soybean Mosaic virus -> soybean mosaic virus
L18 inoculated with SMV strain -> in response to infection of SMV strain SC18
L19 Spell out hpi and dpi.
L21 Spell out GO.
L21-22 Based on the enrichment analyses for gene ontology enrichment and KEGG pathway, we found that 48 dpi was the best time point for the defense response of the two soybean varieties in response to SMV infection.
L24 Expression of seven selected genes were further verified by qRT-PCR.
L25 The sentence sounds strange.
L26 RNA-seq. Glyma.11G239000 (Check space between two sentences).
L28 Expression of genes for XYZ, was downregulated
L29 Expression of Glyma.14G041500 was upregulated
L31 Spell out “SA”.
L40 (L). Merr. (Check the space between words and sentences) There were lots of errors associated with spacing.
L42 I do not believe that SMV is the largest world wide plant virus. It sounds very strange.
L44 in a large area -> worldwide
L54 chromosome 13, 14 and 2, respectively
L69-73 Delete this paragraph describing RSV. This study is about SMV not RSV.
L73-77 Delete this paragraph. Add other studies associated with soybean in response to pathogens.
L91 What does soybean differentials mean?
L105 SC18(rSC18), were (spacing)
L109 method [38].
L109 What is “friction inoculation method”? Please describe it.
L113 Spell out hpi and dpi.
L151-165 To identify differentially expressed genes using DESeq2 or edgeR, the number of mapped reads should be used. Authors indicated that they used FPKM values instead of the number of read counts. Therefore, the results from differentially expressed genes were totally wrong. Please reanalyze all data again.
In Figure 1, authors did not indicate the time after SMV inoculation.
L221 36076218bp -> 36,076,218 bp Please write the number and unit scientifically.
L223 Where is Table S1? Table S1 followed by Table S2. Please check the order of tables.
Figure 2 should be reanalyzed. In addition, the figure quality was very poor.
I cannot see Figure 3, 4, 5, and 6 in detail.
Author Response
RESPONSE TO THE COMMENTS FROM REVIEWER
The authors thank the reviewers very much for kindly providing detailed comments to the manuscript for revision. According to the comments and suggestions, the authors have made a careful revision on the manuscript. The following are the authors' responses to the comments.
Response to reviewer 1
Comment 1: The first was identification of differentially expressed genes. Authors should use the number of mapped reads (read count) instead of FPKM values using DESeq2 or edgeR. Check the manual for two programs.
Response: Yes, in this work, DESeq2 and edgeR were used to analyze the number of mapped reads (read counts). To avoid any misunderstanding, we have corrected the method’s description on ‘Read alignment and differential expression gene screening’ in the revised manuscript.
Comment 2: The second was that the font sizes in the figures were very small. Reanalyze the data and prepare high quality figures for the manuscript.
Response: All figures have been revised and a clear version are provided in the revised manuscript.
Comment 3: The third was that the manuscript was very poorly written. Lots of sentences were awkward. There were lots of mistakes. English was really very bad.
Response: The English language has been polished by an English native speaker.
Comment 4: The fourth was that authors should provide accession numbers of raw data from RNA sequencing after depositing raw data in publica database such as SRA database.
Response: We obtained a total of 155G raw data from RNA sequencing. We are trying hard to upload the raw data to NCBI to meet the deadline when the revised manuscript should be submitted, however, a total of 50G data still need to be uploaded. We will provide the accession numbers later once the uploading process finished.
Other comments:
Q1: The title might be changed to “Comparative transcriptome analyses between resistant and susceptible varieties in response to soybean mosaic virus infection.”
Response: According to the reviewer’s suggestion, the title has been changed to “Comparative transcriptome analyses between resistant and susceptible varieties in response to soybean mosaic virus infection.”
Q2: L3 and L14 Soybean Mosaic virus -> soybean mosaic virus
Response: Corrected.
Q3: L18 inoculated with SMV strain -> in response to infection of SMV strain SC18.
Response: Corrected.
Q4: L19 Spell out hpi and dpi.
Response: They have been changed to “hours post-inoculation and days post-inoculation”.
Q5: L21 Spell out GO.
Response: It has been changed to “gene ontology (GO)”.
Q6: L21-22 Based on the enrichment analyses for gene ontology enrichment and KEGG pathway, we found that 48 dpi was the best time point for the defense response of the two soybean varieties in response to SMV infection.
Response: The sentence “Through the analysis of the GO function of DEGs and the enrichment pathway of KEGG function, it is found that 48 hpi is the best time point for the defense response of Kefeng-1 and NN1138-2 to soybean mosaic virus.” has been changed to “Based on the enrichment analyses for gene ontology enrichment and KEGG pathway, we found that 48 dpi was the best time point for the defense response of the two soybean varieties in response to SMV infection.”
Q7: L24 Expression of seven selected genes were further verified by qRT-PCR.
Response: The sentence “Seven candidate genes selected were further verified by qRT-PCR, the expression profile of which were relatively consistent with the results of RNA-Seq.” has been changed to “Expression of seven candidate genes were further verified by qRT-PCR and were relatively consistent with the results of RNA-Seq.”
Q8: L25 The sentence sounds strange.
Response: The sentence “Glyma.11G239000, Glyma.18G018400 involved in ETH, from ethylene-insensitive3/Ethylene-insensitive3-like (EIN3/EIL) protein family were downregulated in NN1138-2 but not in Kefeng-1” has been changed to “Expression of genes for Glyma.11G239000 and Glyma.18G018400, from ethylene-insensitive 3/Ethylene-insensitive3-like (EIN3/EIL) protein family involving in ETH, were downregulated in NN1138-2 but not in Kefeng-1”.
Q9: L26 RNA-seq. Glyma.11G239000 (Check space between two sentences).
Response: Corrected.
Q10: L28 Expression of genes for XYZ, was downregulated
Response: The sentence “Glyma.11G239000, Glyma.18G018400 involved in ETH, from ethylene-insensitive3/Ethylene-insensitive3-like (EIN3/EIL) protein family were downregulated in NN1138-2 but not in Kefeng-1” has been changed to “Expression of genes for Glyma.11G239000 and Glyma.18G018400, from ethylene-insensitive 3/Ethylene-insensitive3-like (EIN3/EIL) protein family involving in ETH, were downregulated in NN1138-2 but not in Kefeng-1”.
Q11: L29 Expression of Glyma.14G041500 was upregulated
Response: The sentence “Glyma.14G041500 was upregulated.” has been changed to “Expression of Glyma.14G041500 was upregulated.”
Q12: L31 Spell out “SA”.
Response: It has been changed to “Salicylic acid”.
Q13: L40 (L). Merr. (Check the space between words and sentences) There were lots of errors associated with spacing.
Response: Corrected.
Q14: L42 I do not believe that SMV is the largest world wide plant virus. It sounds very strange.
Response: The sentence “Soybean mosaic virus (SMV) is a member of Potyvirus, one of the most broadly distributed viral diseases worldwide in soybean.” has been changed to “Soybean mosaic virus (SMV), a member of the genus Potyvirus, is the major pathogen causing soybean mosaic disease”.
Q15: L44 in a large area -> worldwide
Response: Corrected.
Q16: L54 chromosome 13, 14 and 2, respectively.
Response: “chromosomes13, 14, 2, respectively” has been changed to “chromosomes 13, 14 and 2, respectively”.
Q17: L69-73 Delete this paragraph describing RSV. This study is about SMV not RSV.
Response: Suggestion accepted. The sentence describing RSV has been deleted.
Q18: L73-77 Delete this paragraph. Add other studies associated with soybean in response to pathogens.
Response: This paragraph has been deleted.
Q19: L91 What does soybean differentials mean?
Response: Corrected. Soybean differentials mean soybean identification hosts (a set of soybean varieties with different resistance).
Q20: L105 SC18(rSC18), were (spacing)
Response: Corrected.
Q21: L109 method [38].
Response: Corrected.
Q22: L109 what is “friction inoculation method”? Please describe it.
Response: According to the reviewer’s suggestion, “friction inoculation” has been changed to “mechanical inoculation”. See reference 38 for specific inoculation methods.
Q23: L113 Spell out hpi and dpi.
Response: Corrected.
Q24: L151-165 To identify differentially expressed genes using DESeq2 or edgeR, the number of mapped reads should be used. Authors indicated that they used FPKM values instead of the number of read counts. Therefore, the results from differentially expressed genes were totally wrong. Please reanalyze all data again.
Response: Yes, DESeq2 and edgeR were used to analyze the number of mapped reads (read counts). To avoid any misunderstanding, the sentence “Therefore, the calculated gene expression can be directly used to compare the difference in gene expression among samples. Differential expression analysis of RNAs was performed between two samples different groups using softwares DEGSeq2 and edgeR.” has been changed to “Difference analysis is a statistical analysis of the differences in gene expression between groups. The input data is the reads counts obtained from the gene expression level analysis, which is analyzed by edgeR and DESeq2 software.”
Q25: In Figure 1, authors did not indicate the time after SMV inoculation.
Response: The title of Figure 1 has been changed to “Phenotype of NN 1138-2 and Kefeng-1 inoculated with PBS (left) and SC18 (right) fifteen days later.”
Q26: L221 36076218bp -> 36,076,218 bp Please write the number and unit scientifically.
Response: Corrected.
Q27: L223 Where is Table S1? Table S1 followed by Table S2. Please check the order of tables.
Response: Table S1 is at Line 179 and we have carefully checked the order of all tables in the manuscript.
Q28: Figure 2 should be reanalyzed. In addition, the figure quality was very poor. I cannot see Figure 3, 4, 5, and 6 in detail.
Response: In this work, DESeq2 and edgeR were used to analyze the number of mapped reads (read counts). We have correct the method’s description on ‘Read alignment and differential expression gene screening’ in the revised manuscript. Accordingly, analyzation for Figure 2 was reasonable. In addition, a clear version of Figures 1-6 has been provided separately.

Reviewer 2 Report
- Authors have an emphasis on the role of plant defense hormones (ET, JA, and SA) in resistant and susceptible soybean varieties. I think they also need to perform real-time real-time PCR experiment with some marker genes of ET, JA, and SA pathways.
- They have good data at Q30, then why then analyze their work at Q20?
- There are several typographical mistakes
- Gene's name should be in italic.
- Picture quality must be improved.
Author Response
RESPONSE TO THE COMMENTS FROM REVIEWER
The authors thank the reviewers very much for kindly providing detailed comments to the manuscript for revision. According to the comments and suggestions, the authors have made a careful revision on the manuscript. The following are the authors' responses to the comments.
Response to reviewer 2
Q1: Authors have an emphasis on the role of plant defense hormones (ET, JA, and SA) in resistant and susceptible soybean varieties. I think they also need to perform real-time PCR experiment with some marker genes of ET, JA, and SA pathways.
Response: Thanks for your good suggestion, and we have already performed this experiment, following we will further explore the molecular function of those key genes of ET, JA, and SA pathways during soybean and SMV interaction.
Q2: They have good data at Q30, then why then analyze their work at Q20?
Response: The sentence “The Q20 percentages of the raw data were all above 97.25%.” has been changed to “The Q30 percentages of the raw data were all above 92.32%.
Q3: There are several typographical mistakes
Response: We have carefully checked the manuscript and corrected the typographical mistakes.
Q4: Gene's name should be in italic.
Response: Corrected. eg. Glyma.03g28650, Glyma.19g31395 and Glyma.11g33790, and the like that.
Q5: Picture quality must be improved.
Response: A clear version of Figures 1-6 has been provided again.

Reviewer 3 Report
- Line 19: what are hpi and dpi?
- Line 26: What are Glyma.11G239000 and Glyma.18G018400? You should be more specific
- Line 55: Do not start a sentence with an abbreviation
- The abstract is full of abbreviations that do not ease the reading process.
- Line 114 – 115: Why do you say: samples shall be frozen? Does that mean some samples were not frozen? Rewrite for clarity
- Section 2.1. Did you collect inoculated samples without symptoms? Why do you need to confirm disease symptoms on non-collected and inoculated plants?
- Section 2.2. What was the RIN number of total RNA? What was the cDNA library size?
- Line 148: Provide a reference for HISAT
- Lines 171 – 172: Provide the date when the database was accessed.
- Figures 2, and 3 are blurred
4. Discussion section. Provide a putative scheme of the resistance process to SMV strain SC18 via ETH, JA, SA signaling pathways.
Author Response
RESPONSE TO THE COMMENTS FROM REVIEWER
The authors thank the reviewers very much for kindly providing detailed comments to the manuscript for revision. According to the comments and suggestions, the authors have made a careful revision on the manuscript. The following are the authors' responses to the comments.
Response to reviewer 3
Q1: - Line 19: what are hpi and dpi?
Response: They have been changed to “hours post-inoculation and days post-inoculation”.
Q2: - Line 26: What are Glyma.11G239000 and Glyma.18G018400? You should be more specific
Response: Glyma.11G239000 and Glyma.18G018400 are members of ethylene-insensitive3/Ethylene-insensitive3-like (EIN3/EIL) protein family which involved in ETH pathway. Their detailed information was described in Table 1.
Q3: Line 55: Do not start a sentence with an abbreviation. The abstract is full of abbreviations that do not ease the reading process.
Response: Corrected. The abbreviations have been spelled out in the revised manuscript.
Q4: - Line 114 – 115: Why do you say: samples shall be frozen? Does that mean some samples were not frozen? Rewrite for clarity
Response: Sorry for the misunderstanding. This sentence has been corrected to “All samples collected at each time-point were frozen immediately in liquid nitrogen” .
Q5: - Section 2.1. Did you collect inoculated samples without symptoms? Why do you need to confirm disease symptoms on non-collected and inoculated plants?
Response: Yes. SMV symptoms were usually appeared at 7 to 10 days post inoculation. In this work, leaves were sampled at 0, 6, 48 hours post inoculation and 5 days post inoculation. To ensure the inoculated plants were successfully infected with SMV, the remaining plants were grown in the growth chamber until the symptoms appeared. To avoid any misunderstanding, the Section 2.1 has been revised.
Q6: - Section 2.2. What was the RIN number of total RNA? What was the cDNA library size?
Response: The RNA integrity value was around 7. The size of the cDNA library is 300-400 bp.
Q7: - Line 148: Provide a reference for HISAT
Response: The reference has been provided.
Q8: - Lines 171-172: Provide the date when the database was accessed.
Response: Here, the database website address is provided in the manuscript. (https://soybase.org/gb2/gbrowse/gmax2.0/). The database was accessed on July 24, 2018.
Q9: - Figures 2, and 3 are blurred
Response: A clear version of Figures 1-6 has been provided.
Q10: 4. Discussion section. Provide a putative scheme of the resistance process to SMV strain SC18 via ETH, JA, SA signaling pathways.
Response: We further need to perform a real-time PCR experiment with some marker genes of ET, JA, and SA pathways.

Round 2
Reviewer 1 Report
Authors revised the manuscript properly according to reviewer's comments.
However, the raw data from RNA-sequencing were not deposited in the public database such as SRA database.
It is necessary to deposit all raw data in SRA database and make them in public for the acceptance of your manuscript.
It will not take long time as authors think.
Yesterday, I also deposit 24 fastq files in SRA database.
It took less than two hours.
Go to the SRA database. Register using google ID. Register project for the SRA data. After that, you can register biosamples and upload all raw data via aspera program.
Few hours later, you will get accession numbers for each raw data and authors provide these accession numbers in the manuscript.
Author Response
RESPONSE TO THE COMMENT FROM REVIEWER 1
The authors thank the reviewer very much for kindly providing detailed comments to the manuscript for revision. The following is the authors' response to the comment.
Response to reviewer 1
Comment 1: Authors revised the manuscript properly according to reviewer's comments.
However, the raw data from RNA-sequencing were not deposited in the public database such as SRA database.
It is necessary to deposit all raw data in SRA database and make them in public for the acceptance of your manuscript.
It will not take long time as authors think.
Yesterday, I also deposit 24 fastq files in SRA database.
It took less than two hours.
Go to the SRA database. Register using google ID. Register project for the SRA data. After that, you can register biosamples and upload all raw data via aspera program.
Few hours later, you will get accession numbers for each raw data and authors provide these accession numbers in the manuscript.
Response: The authors thank the reviewer for your detailed guidance, the raw data from RNA-sequencing have been deposited in the public database NCBI and the accession number is PRJNA861407 (https://www.ncbi.nlm.nih.gov/sra/PRJNA861407). The accession number was also provided in the revised manuscript.
